# NeMo-map: Neural Implicit Flow Fields for Spatio-Temporal Motion Mapping

**Yufei Zhu**[1]* **Shih-Min Yang**[1] **Andrey Rudenko**[2] **Tomasz P. Kucner**[3]
**Achim J. Lilienthal**[1,2] **Martin Magnusson**[1]

[1]Örebro University, Sweden    [2]Technical University of Munich, Germany
[3]Aalto University, Finland

## Abstract

Safe and efficient robot operation in complex human environments can benefit from good models of site-specific motion patterns. Maps of Dynamics (MoDs) provide such models by encoding statistical motion patterns in a map, but existing representations use discrete spatial sampling and typically require costly offline construction. We propose a continuous spatio-temporal MoD representation based on implicit neural functions that directly map coordinates to the parameters of a Semi-Wrapped Gaussian Mixture Model. This removes the need for discretization and imputation for unevenly sampled regions, enabling smooth generalization across both space and time. Evaluated on two public datasets with real-world people tracking data, our method achieves better accuracy of motion representation and smoother velocity distributions in sparse regions while still being computationally efficient, compared to available baselines. The proposed approach demonstrates a powerful and efficient way of modeling complex human motion patterns and high performance in the trajectory prediction downstream task. The code is publicly available at `https://github.com/test-bai-cpu/nemo-map`.

## 1 Introduction

Safe and efficient operation in complex, dynamic and densely crowded human environments is a critical prerequisite for deploying robots in various tasks to support people in their daily activities. Extending the environment model with human motion patterns using a *map of dynamics* (MoD) is one way to achieve unobtrusive navigation, compliant with existing site-specific motion flows (Palmieri et al., 2017; Swaminathan et al., 2022).

Incorporating MoDs into motion planning provides benefits in crowded environments, since they encode information about the expected motion outside of the robot's sensor range, allowing for less reactive behavior. MoDs can also be applied to long-term human motion prediction (Zhu et al., 2024; 2025a). MoDs help predict realistic trajectories that implicitly respect the complex topology of the environment, such as navigating around corners or avoiding obstacles.

Several approaches have been proposed for constructing MoDs. Early methods modeled human motion on occupancy grid maps, treating dynamics as shifts in occupancy (Wang et al., 2015; 2016). These approaches struggle with noisy or incomplete trajectory data. Later, velocity-based representations have been introduced, most notably the CLiFF-map (Kucner et al., 2017), which models local motion patterns with Gaussian mixture models, effectively captures multimodality in human flows and has been successfully used in both robot navigation and prediction tasks. The methods above are computed in batch, given a set of observations. Online learning methods have also been explored to update motion models as new observations arrive (Zhu et al., 2025b), allowing robots to adapt to changing environments without costly retraining from scratch. Temporal MoDs have also been explored, including STeF-maps (Molina et al., 2022), which apply frequency-based models to encode periodic variations in the flow.

---

*Correspondence to: `yufei.zhu@oru.se`

However, existing MoDs require spatial discretization, with a manually selected map resolution for point locations and interpolation to estimate motion at arbitrary positions. This discretization introduces information loss, reduces flexibility, and complicates tuning across different environments.

To address these challenges, in this work, instead of representing motion patterns on a discrete grid, we propose a *continuous map of dynamics* using implicit neural representation. We learn a neural function that maps *spatio-temporal coordinates* to parameters of a local motion distribution. Implicit neural representations have emerged as powerful tools for encoding continuous functions, providing compact and differentiable models with strong generalization. Leveraging these properties, this formulation allows the model to smoothly generalize across space and time, while maintaining multimodality in places where flows tend to go in more than one direction since it produces a wrapped Gaussian mixture model of expected motion given a query location and time.

We evaluate our approach on real-world datasets and show that continuous MoDs not only improve representation accuracy but can also be computationally efficient. Our method yields smoother and more consistent velocity distributions, resulting in more accurate representations of human motion patterns and higher performance in the trajectory prediction downstream task. In contrast to baseline approaches that rely on time-consuming per-cell motion modeling, it trains a model that represents motion continuously over space and time and enables fast inference at arbitrary locations. Unlike the faster but discretised representation of STeF-map Molina et al. (2022), our method preserves non-discretised directions, yielding results closer in spirit to CLiFF.

In summary, the main contribution of this work is an entirely novel representation of flow-aware maps of dynamics, named NeMo-map. In contrast to existing methods, NeMo allow *continuous spatio-temporal queries* to generate location- and time-specific *multimodal flow predictions*. As evidenced by our experimental validation on real-world human motion data, NeMo efficiently learns a highly accurate statistical representation of motion in large-scale maps.

## 2 RELATED WORK

A *map of dynamics* (MoD) is a representation that augments the geometric map of an environment with statistical information about observed motion patterns. Unlike static maps, MoDs incorporate spatio-temporal flow information, allowing robots to reason about how humans typically move in a given environment.

MoDs can be built from various sources of input, such as trajectories (Bennewitz et al., 2005), dynamics samples, or information about the flow of continuous media (e.g., air or water) (Bennetts et al., 2017). Furthermore, these models can feature diverse underlying representations, including evidence grids, histograms, graphs, or Gaussian mixtures.

Several MoDs types are described in the literature, providing an efficient tool for storing and querying about historical or expected changes in environment states. Occupancy-based methods focus on mapping dynamics on occupancy grids, modeling motion as shifts in occupancy (Wang et al., 2015; 2016). Trajectory-based methods extract trajectories and group into clusters, with each cluster representing a typical path through the environment (Bennewitz et al., 2005). These approaches suffer from noisy or incomplete trajectories. To address this, Chen et al. (2016) formulates trajectory modeling as a dictionary learning problem and uses augmented semi-nonnegative sparse coding to find motion patterns characterized by partial trajectory segments.

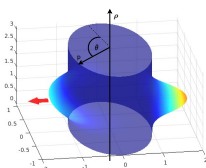

Figure 1: Probability density of a Semi-Wrapped Gaussian Mixture Model (SWGMM) with two components, visualized on a cylinder. Orientation $\theta$ is wrapped around the circular axis, while speed $\rho$ extends along the vertical axis. The representation allows joint modeling of angular (orientation) and linear (speed) variables, capturing multimodality in motion patterns.

MoDs can also be based on velocity observations. With velocity mapping, human dynamics can be modeled through flow models. Kucner et al. (2017) presented a probabilistic framework for mapping velocity observations, named Circular-

Linear Flow Field map (CLiFF-map). CLiFF-map represents local flow patterns as a multi-modal, continuous joint distribution of speed and orientation, as further described in Sec. 3. A benefit of CLiFF-map is that it can be built from incomplete or spatially sparse velocity observations (de Almeida et al., 2024), without the need to store a long history of data or deploy advanced tracking algorithms. CLiFF-maps are typically built offline, for the reason of high computational costs associated with the building process. This constraint limits their applicability in real environments.

When building flow models, temporal information can also be incorporated. Molina et al. (2022) apply the Frequency Map Enhancement (FreMEn Krajník et al. (2017)), which is a model describing spatio-temporal dynamics in the frequency domain, to build a time-dependent probabilistic map to model periodic changes in people flow called STeF-map. The motion orientations in STeF-map are discretized. Another method of incorporating temporal information is proposed by Zhi et al. (2019). Their approach uses a kernel recurrent mixture density network to provide a multimodal probability distribution of movement directions of a typical object in the environment over time, though it models only orientation and not the speed of human motion.

It is important to note that a map of dynamics is not a trajectory prediction model. Whereas trajectory predictors (e.g., LSTMs) aim to forecast the future state of agents by propagating state information forward in time from an initial state, our goal is fundamentally different. We seek to construct a spatio-temporal prior that encodes the distribution of motion patterns in the environment itself. This prior can be queried directly at any spatial coordinate and any time of day, providing motion statistics that can support downstream tasks such as planning or long-term prediction, but it does not by itself generate trajectories for individual agents.

## 3 METHODOLOGY

### 3.1 PROBABILISTIC MODELING OF HUMAN MOTION

Our spatio-temporal map of dynamics produces probability distributions over human motion velocities. A velocity $\mathbf{v}$ is defined by the pair of *speed* (a positive linear variable $\rho \in \mathbb{R}^+$) and *orientation* (a circular variable $\theta \in [0, 2\pi)$).

To capture the statistical structure of such data, we model human motion patterns with a *Semi-Wrapped Gaussian Mixture Model* (SWGMM), similar to the CLiFF-map representation (Kucner et al., 2017). While a von Mises distribution would be effective for purely angular variables, is not suitable when combining circular and linear components. Roy et al. (2012) proposed the von Mises-Gaussian mixture model (VMGMM) to jointly represent one circular variable and linear variables. However, their model assumes independence between the circular and linear dimensions, which limits its ability in capturing real-world correlations. To overcome this, SWGMM (Roy et al., 2016) jointly models circular-linear variables and allows correlations between them.

An SWGMM models velocity $\mathbf{v} = [\rho, \theta]^\top$ as a mixture of $J$ Semi-Wrapped Normal Distributions (SWNDs):

$$p(\mathbf{v} \mid \boldsymbol{\xi}) = \sum_{j=1}^{J} w_j \mathcal{N}_{\boldsymbol{\mu}_j, \boldsymbol{\Sigma}_j}^{\text{SW}}(\mathbf{v}), \tag{1}$$

where $\boldsymbol{\xi} = \{\xi_j = (w_j, \boldsymbol{\mu}_j, \boldsymbol{\Sigma}_j)\}_{j=1}^{J}$ denotes a finite set of components of the SWGMM. Each $w_j$ is a mixing weight and satisfies $0 \leq w_j \leq 1)$, $\boldsymbol{\mu}_j$ the component mean, and $\boldsymbol{\Sigma}_j$ the covariance. An SWND $\mathcal{N}_{\boldsymbol{\Sigma}, \boldsymbol{\mu}}^{\text{SW}}$ is formally defined as

$$\mathcal{N}_{\boldsymbol{\Sigma}, \boldsymbol{\mu}}^{\text{SW}}(\mathbf{v}) = \sum_{k \in \mathbb{Z}} \mathcal{N}_{\boldsymbol{\mu}, \boldsymbol{\Sigma}}\left([\rho, \theta]^\top + 2\pi[0, k]^\top\right), \tag{2}$$

where $k$ is a winding number. In practice, the PDF can be approximated adequately by taking $k \in \{-1, 0, 1\}$ (Mardia & Jupp, 2008).

The SWGMM density function over velocities can be visualized as a function on a cylinder, as shown in Fig. 1. Orientation values $\theta$ are wrapped on the unit circle and the speed $\rho$ runs along the length of the cylinder. This formulation yields a flexible and interpretable probabilistic representation of local human motion, capturing multimodality and correlations between orientation and speed.

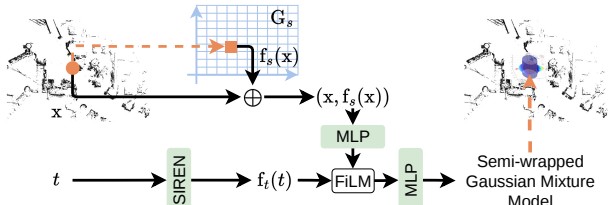

Figure 2: Method overview. A spatio-temporal query $(\mathbf{x}, t)$ is mapped to parameters of a Semi-Wrapped Gaussian Mixture Model (SWGMM). The spatial coordinate $\mathbf{x}$ is used to interpolate features from a learnable spatial grid $\mathbf{G}_s$, and the temporal coordinate $t$ is encoded using a SIREN network. The spatial features $\mathbf{f}_s(\mathbf{x})$, temporal encoding $\mathbf{f}_t(t)$, and raw coordinates are fused via FiLM (Perez et al., 2018) and passed through an MLP, which outputs the parameters of a SWGMM, providing a continuous, multimodal probabilistic representation of motion dynamics at the queried location and time.

## 3.2 LEARNING CONTINUOUS MOTION FIELDS

Previous MoD approaches, such as CLiFF-maps and STeF-maps, rely on discretizing the environment into cells and fitting local probability models. Discretization leads to information loss and prevents querying at arbitrary locations. We address this by introducing a *continuous* map of dynamics parameterized by a *neural implicit* representation. The method overview is shown in Fig. 2.

To capture temporal variations in human motion, we condition the MoD on a continuous temporal variable. Incorporating time into the input enables the model to capture temporal variations without requiring sequential rollouts, and enables efficient queries of motion dynamics at arbitrary spatio-temporal coordinates $(x, y, t)$.

**Problem statement.** Given a dataset $\mathcal{D}$ of $\mathcal{N}$ spatio-temporal motion samples:

$$\mathcal{D} = \{(\mathbf{x}_i, t_i, \mathbf{v}_i)\}_{i=1}^{N},$$

where $\mathbf{x}_i \in \mathbb{R}^2$ is the spatial coordinate, $\mathbf{t}_i \in \mathbb{R}$ is a continuous temporal variable, and $\mathbf{v}_i = [\rho_i, \theta_i]^\top$ is observed velocity, we learn a continuous function $\Phi_\theta$ that maps a spatio-temporal coordinate $(\mathbf{x}, t)$ to SWGMM parameters:

$$\Phi_\theta(\mathbf{x}, t) = \left\{ w_j(\mathbf{x}, t), \, \boldsymbol{\mu}_j(\mathbf{x}, t), \, \boldsymbol{\Sigma}_j(\mathbf{x}, t) \right\}_{j=1}^{J}, \tag{3}$$

where $J$ is the number of mixture components, weights $w_j \geq 0$ and $\sum_{j=1}^{J} w_j = 1$. Each of the $j$ components models the joint velocity $\mathbf{v} = [\rho, \theta]^\top$ with a Semi-Wrapped Normal Distribution $\mathcal{N}^{\mathrm{SW}}_{\boldsymbol{\Sigma}, \boldsymbol{\mu}}$. At inference time, querying $\Phi_\theta$ at any coordinate yields the full set of SWGMM parameters, resulting in a continuous probabilistic representation of motion dynamics. This formulation enables the model to learn smooth, continuous motion fields while retaining the multimodal characteristic of human motion.

**Architecture.** In our neural representation, we parameterize $\Phi_\theta$ with a fully connected multilayer perceptron (MLP), conditioned on both spatial and temporal features:

$$\underbrace{\mathbf{f}_s(\mathbf{x})}_{\text{spatial features}} \in \mathbb{R}^{C_s}, \qquad \underbrace{\mathbf{f}_t(t)}_{\text{temporal encoding}} \in \mathbb{R}^{C_t}.$$

For spatial features, a learnable grid $\mathbf{G}_s \in \mathbb{R}^{H \times W \times C_s}$ is queried at location $\mathbf{x}$ by bilinear interpolation, producing $\mathbf{f}_s(\mathbf{x})$. This captures local variations in motion patterns while remaining continuous in space.

For temporal encoding, we encode $t$ with SIREN, the sinusoidal representation network (Sitzmann et al., 2020), which uses periodic activation functions throughout the network.

The MLP input concatenates the raw coordinates and the spatial and temporal features, $\mathbf{z} = [\mathbf{x}, t, \mathbf{f}_s(\mathbf{x}), \mathbf{f}_t(t)]$, and outputs SWGMM parameters. This feature-conditioned representation enables the model to flexibly encode local variations in motion dynamics while maintaining global smoothness across both space and time.

**Likelihood and training.** For a spatio-temporal coordinate $(\mathbf{x}_i, t_i)$, the velocity likelihood under the predicted SWGMM is

$$p(\mathbf{v}_i \mid \Phi_\theta(\mathbf{x}_i, t_i)) = \sum_{j=1}^{J} w_j(\mathbf{x}_i, t_i) \, \mathcal{N}^{\text{SW}}_{\boldsymbol{\mu}_j(\mathbf{x}_i, t_i), \, \boldsymbol{\Sigma}_j(\mathbf{x}_i, t_i)}(\mathbf{v}_i),$$

where $\mathcal{N}^{\text{SW}}$ denotes the semi-wrapped normal distribution that wraps the angular component (see Eq. (2)). The model is trained by minimizing the negative log-likelihood of motion samples from the dataset under the probability density function (PDF) produced by the model:

$$\mathcal{L}(\theta) = -\frac{1}{N} \sum_{i=1}^{N} \log p\big(\mathbf{v}_i \mid \Phi_\theta(\mathbf{x}_i, t_i)\big).$$

## 4 RESULTS

To evaluate spatio-temporal maps of dynamics that capture changes of human motion patterns over time, it is essential to use datasets that span multiple days and reflect variations in human motion patterns. Our experiments were conducted using two real-world datasets, ATC (Brščić et al., 2013) and ETH (Pellegrini et al., 2009)/UCY (Lerner et al., 2007). The ATC dataset provides sufficient multi-day coverage for evaluation, consisting of long-term real-world people tracking data collected in a shopping mall. The ETH/UCY dataset contains pedestrian trajectories captured in outdoor environments such as university campuses, with multiple unique scenes. Details are in Appendix B.

### 4.1 BASELINES

**Circular-Linear Flow Field Map (CLiFF-map).** CLiFF-map (Kucner et al., 2017) represents motion patterns by associating each discretized grid location with an SWGMM fitted from local observations. The environment is divided into a set of grid locations, and each grid location aggregates motion samples within a fixed radius. The SWGMM parameters at each grid location are estimated via expectation-maximization (EM) (Dempster et al., 1977), with the number and initial positions of mixture components determined using mean shift clustering (Cheng, 1995). When training the CLiFF-map, the convergence precision is set to 1e–5 for both mean shift and EM algorithms, with a maximum iteration count of 100. The grid resolution is set to $1\,\text{m}$. To evaluate different hours, we train separate CLiFF-maps for each hour using the motion samples observed during that time.

**Online CLiFF-map.** Online CLiFF-map (Zhu et al. (2025b)) extends the static CLiFF model by updating the SWGMM parameters incrementally as new motion observations become available. Each grid location maintains an SWGMM, which is initialized upon first receiving observations and subsequently updated using the stochastic expectation-maximization (sEM) algorithm (Cappé & Moulines (2009)). In sEM, the expectation step of the original EM algorithm is replaced by a stochastic approximation step, while the maximization step remains unchanged. Like the static CLiFF-map, Online CLiFF-map outputs SWGMM parameters at each grid location, but supports continuous adaptation over time. In the experiments, we follow a spatio-temporal setting by generating an online CLiFF-map for each hour. Observations collected in an hour interval are treated as the new data batch for updating SWGMMs, producing a temporally adaptive motion representation.

**Spatio-Temporal Flow Map (STeF-map).** STeF-map (Molina et al., 2022) is a spatio-temporal map of dynamics that models the likelihood of human motion directions using harmonic functions. Each grid location maintains $k_{\text{stef}}$ temporal models, corresponding to $k_{\text{stef}}$ discretized orientations of people moving through that location over time. By modeling periodic patterns, STeF-map captures long-term temporal variations in crowd movements and can predict activities at specific times of day under the assumption of periodicity in the environment. Following Molina et al. (2022), we set $k_{\text{stef}} = 8$ in the experiments, and the model orders for training STeF-map, i.e. the number of periodicities, is set to 2.

### 4.2 QUANTITATIVE RESULTS

To quantitatively evaluate the accuracy of modeling human motion patterns (MoD quality), we use the negative log-likelihood (NLL). An MoD represents human motion as a probability distribution over velocity conditioned on a spatio-temporal coordinate $(x, y, t)$, implemented as either an

SWGMM (our method and CLiFF-maps) or a histogram (STeF-map). To evaluate representation accuracy, we use test data consisting of observed human motions in the same environment. For each test sample $(x, y, t)$, we query the MoD to obtain the predicted distribution and compute the likelihood of the observed motion under this distribution. A higher likelihood indicates that the predicted distribution better aligns with the observed data. We report NLL for numerical stability and easy comparison, so lower NLL values correspond to more accurate motion representations, i.e., higher quality MoDs. Table 1 and Table 2 report the accuracy results for the ATC dataset and the ETH/UCY dataset, respectively. Our method achieves the lowest NLL, outperforming all baselines. Online CLiFF-map, CLiFF-map, and STeF-map exhibit higher NLLs, with paired t-tests showing $p < 0.001$ under the null hypothesis that baseline NLL is less than or equal to ours.

Compared with STeF-maps, methods based on SWGMM, such as ours and CLiFF-map, offer two key advantages. They jointly model speed and orientation, whereas STeF-maps do not include speed information. In addition, SWGMMs represent orientation continuously rather than through a discretized 8-bin histogram as in STeF-map. These aspects lead to a more accurate representation of human motion and contribute to the improved performance.

Limitations of CLiFF-maps are from discretizing the environment into grid cells, with each cell storing a locally fitted SWGMM. This grid-based design limits spatial resolution and introduces discontinuities at cell boundaries in both space and time. In particular, dividing time into hourly intervals is a coarse approximation that can produce abrupt changes, since human motion patterns do not necessarily shift at exact hour boundaries. In contrast, our method models the MoD as a continuous neural implicit representation. This enables smooth generalization across space and time, supports queries at arbitrary spatio-temporal coordinates, and provides a compact representation that avoids the memory overhead of storing distributions for every grid cell.

We also compare the map building time as shown in Table 3. For the baselines, the training time corresponds to convergence on all grid cells, while for our method it corresponds to the neural network training time. Our method is computationally efficient, training on a full day of data in under 20 minutes, while achieving higher accuracy. These results highlight the practicality of continuous MoDs for real-time applications, combining both accuracy and efficiency.

Table 1: Accuracy evaluation on the ATC dataset using average negative log-likelihood (NLL), where lower values indicate better accuracy. We report mean $\pm$ standard deviation, together with the reduction in NLL relative to our method and the corresponding 95% confidence interval (CI).

| Method | NLL$\downarrow$ | NLL reduction (vs Ours) | 95% CI |
|---|---|---|---|
| **Ours** | **0.775 $\pm$ 2.052** | – | – |
| Online CLiFF-map | 1.527 $\pm$ 4.156 | +0.752 | [0.749, 0.755] |
| CLiFF-map | 1.964 $\pm$ 4.953 | +1.189 | [1.185, 1.192] |
| STeF-map | 5.576 $\pm$ 9.314 | +4.801 | [4.794, 4.809] |

Table 2: Accuracy evaluation on the ETH/UCY dataset using NLL, where lower values indicate better accuracy. We report mean $\pm$ standard deviation, together with the reduction in NLL relative to our method and the corresponding 95% confidence interval (CI) of this reduction.

| Method | ETH | HOTEL | UNIV | ZARA |
|---|---|---|---|---|
| **Ours** | **-0.384 $\pm$ 2.051** | **-0.838 $\pm$ 4.043** | **0.404 $\pm$ 1.902** | **-0.342 $\pm$ 2.152** |
| CLiFF-map | 0.112 $\pm$ 4.005 | 0.701 $\pm$ 4.533 | 0.518 $\pm$ 2.125 | 0.068 $\pm$ 4.265 |
| Reduction vs Ours | +0.496 | +1.539 | +0.114 | +0.410 |
| 95% CI | [0.344, 0.648] | [1.287, 1.791] | [0.049, 0.178] | [0.215, 0.604] |
| Online CLiFF-map | 0.086 $\pm$ 4.451 | 1.241 $\pm$ 7.142 | 0.577 $\pm$ 2.548 | 0.186 $\pm$ 5.477 |
| Reduction vs Ours | +0.470 | +2.079 | +0.173 | +0.528 |
| 95% CI | [0.307, 0.633] | [1.744, 2.412] | [0.098, 0.247] | [0.264, 0.792] |
| STeF-map | 2.315 $\pm$ 6.016 | 3.349 $\pm$ 7.660 | 10.932 $\pm$ 12.771 | 2.784 $\pm$ 7.014 |
| Reduction vs Ours | +2.699 | +4.187 | +10.528 | +3.126 |
| 95% CI | [2.472, 2.926] | [3.820, 4.554] | [10.187, 10.868] | [2.779, 3.472] |

Table 3: Training and inference times for map building on the ATC dataset. Lower values indicate faster performance. Experiments were conducted on a desktop computer equipped with an Intel i9-12900K CPU and an NVIDIA GeForce RTX 3060 GPU running Ubuntu 20.04.

| Method | Train time (minute)↓ | Inference time (second)↓ |
|---|---|---|
| Ours | 19.26 | $1.363 \times 10^{-6}$ |
| Online CLiFF-map | 23.859 | $1.914 \times 10^{-3}$ |
| CLiFF-map | 1831 | $1.914 \times 10^{-3}$ |
| STeF-map | 0.815 | $5.665 \times 10^{-5}$ |

## 4.3 QUALITATIVE RESULTS

Fig. 3 and Fig. 4 present qualitative examples of NeMo-map compared with baseline MoDs on the ATC dataset. NeMo-map captures multimodal human motion patterns, which is important in open spaces with intersecting flows. Fig. 3 presents the central open area of the ATC shopping mall. In the region highlighted by a black circle, a dominant horizontal flow intersects with a vertical flow. NeMo-map preserves both motion modes, maintaining the full continuity of the crossing traffic. In contrast, CLiFF-map fails to capture this multimodality consistently. It shows the vertical flow only at the top and bottom of the marked region, while cutting off the middle section, resulting in an incomplete representation of the crossing motion. As a result, NeMo-map achieves lower NLL values and provides accurate representation, as shown in the NLL heatmaps in the bottom row.

Fig. 4 shows the NeMo-map and baseline MoDs, together with corresponding NLL heatmaps, in the east corridor of the ATC shopping mall across different times of the day. In this corridor (right side of the ATC map), human motion patterns vary throughout the day. The dominant flow is directed left/upwards in the morning, shifts direction around noon, and turns into right/downwards in the evening. The flow fields generated by NeMo-map capture such temporal variations and implicitly align with the environment's topology, even though no explicit map was provided during training. For instance, pedestrian speeds decrease near resting benches, motion flows pass through exits, and flows follow the corridors. Same to the middle-region observations in Fig. 3, NeMo-map not only captures the dominant following flow but also preserves the flow that crosses the corridor, which baseline MoDs fail to represent consistently. The richer representation leads to lower NLL values for NeMo-map, demonstrating its higher accuracy in modeling time-varying human dynamics.

Fig. 5 presents generated MoDs in ETH/UCY datasets. The first row shows the ZARA scene at frame 600, along with the pedestrian trajectories for reference. The dominant motion trend in ZARA is along the horizontal direction. While baseline MoDs mostly conform to this dominant flow, NeMo-map adapts to an emerging upper-left directional trend. The second row shows NeMo-map results for three other ETH/UCY scenes: ETH, HOTEL, and UNIV. Pedestrian trajectories from the entire sequences are shown. NeMo-map captures diverse motion patterns. For example in the HOTEL scene, pedestrians tend to keep right when encountering oncoming flows.

## 4.4 DOWNSTREAM TASK VALIDATION

We evaluate NeMo-map in the downstream task of long-term human motion prediction (LHMP). Accurate LHMP is essential for various applications, include optimized motion planning, advanced automated driving, and improved human-robot interaction. While short-term predictions can often rely on current state and immediate interactions, long-term predictions demand more comprehensive modeling of human motion patterns. Maps of Dynamics (MoDs) provide a way to encode such patterns and can act as a prior that guides motion prediction.

Experiments of LHMP are conducted on ATC using the same testing split as in Sec. 4.2. The prediction horizon is up to $60\,\text{s}$. We compare NeMo-map with MoD-based baselines: CLiFF-LHMP (Zhu et al., 2023) and STeF-LHMP (Molina et al., 2022), a diffusion-based model (Gu et al., 2022) and a transformer-based model (Shi et al., 2023). Implementation details and evaluation metrics are provided in Appendix D.

Table 4 presents LHMP performance on the ATC dataset with the prediction horizon of $60\,\text{s}$. Results for other prediction horizons from $10\,\text{s}$ to $60\,\text{s}$ are detailed in Fig. 6. NeMo-map achieves the best

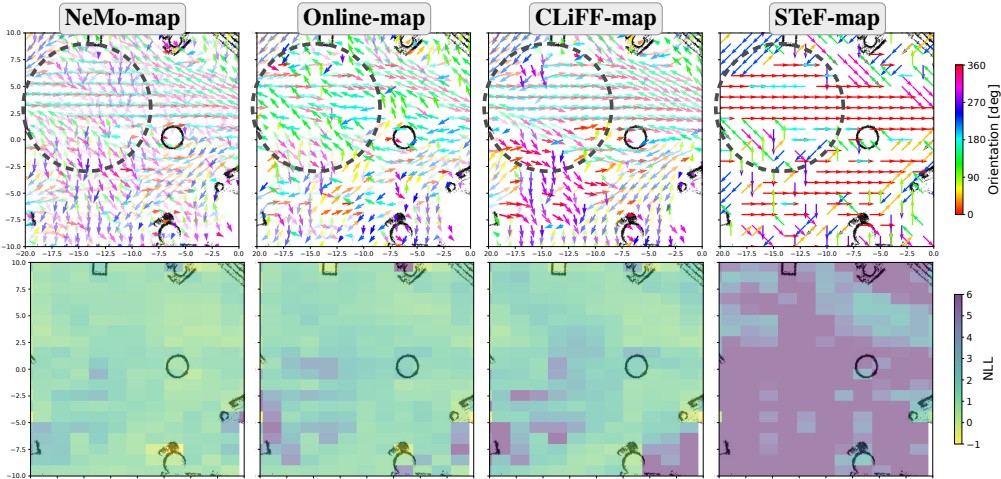

Figure 3: NeMo-map and baseline MoDs in the central open area of the ATC shopping mall. NeMo-map is continuous in space, but for visualization, we query it on a uniform grid, while baseline MoDs are defined only on discrete grid cells. The **top** row shows the velocity distributions generated by each MoD. At each location, arrow color encodes motion orientation and arrow length encodes speed. The black circle highlights a region where horizontal and vertical flows intersect. NeMo-map preserves both motion modes, while baseline MoDs break the vertical flow continuity. The **bottom** row shows the corresponding NLL heatmaps, where lighter values indicate lower NLL, i.e., more accurate representations.

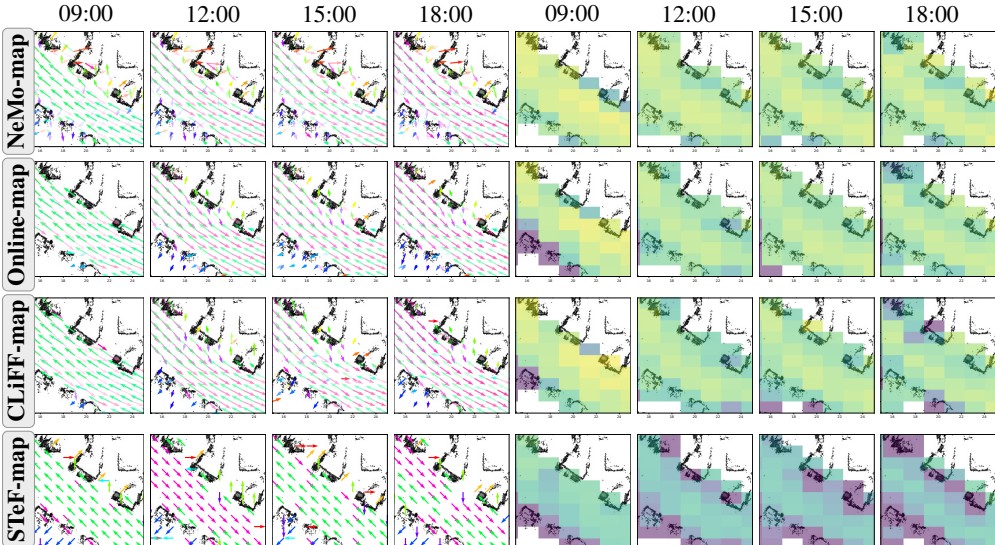

Figure 4: NeMo-map and baseline MoDs in the east corridor, located on the right side of the ATC map (Fig. 7), at different times of the day. The **left** four columns show the generated velocity fields. Nemo-map captures the temporal variation of pedestrian flows. The **right** four columns show the corresponding NLL heatmaps. Compared with the baselines, NeMo-map preserves both the dominant corridor following flow and the crossing flow, leading to consistently lower NLL values.

performance among MoD-based predictors, with paired t-tests showing $p < 0.01$, under the null hypothesis that our method yields equal or higher mean error than the baseline.

The improvements arise from NeMo-map's ability to provide smooth and accurate motion fields across continuous space and time, in contrast to CLiFF-LHMP, which relies on a grid of SWGMMs and search for nearby cells during prediction. By modeling the velocity distribution continuously, NeMo-map captures more precise local flow patterns that result in more accurate long-term predictions. In addition, unlike STeF-map which represents orientation using an 8-bin histogram and

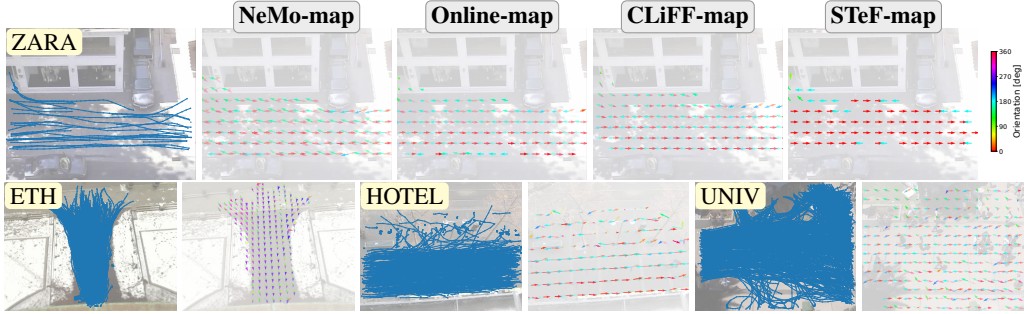

Figure 5: **Top row**: NeMo-map and baseline MoDs in the ZARA scene at frame 600. Pedestrian trajectories are shown as reference. NeMo-map adapts to the changes from dominant horizontal flow to an emerging flow toward the upper-left entrance, while baseline MoDs remain aligned with the horizontal movement. **Bottom row**: NeMo-map results for the ETH, HOTEL and UNIV scenes with trajectories from the entire sequences. It captures multimodal human motion dynamics across different environments.

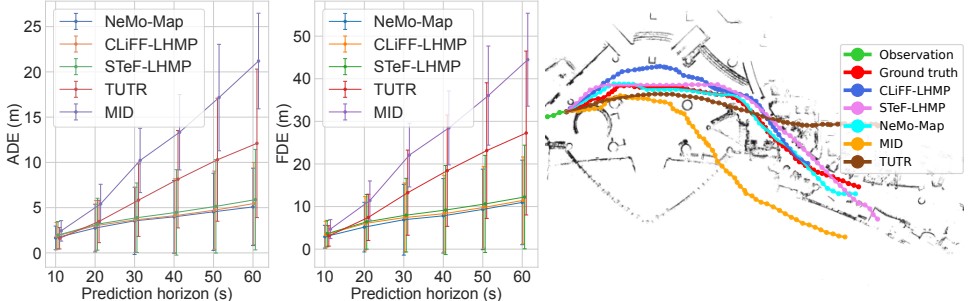

Figure 6: **Left**: ADE/FDE (mean $\pm$ one std. dev.) in the ATC dataset with planning horizon 10–60 s. **Right**: Human motion prediction with a 60 s horizon. The **red** line represents the ground truth trajectory and the **green** line represents the observed trajectory. Prediction with NeMo-map achieves more accurate and realistic predictions than baseline methods. While the trajectories predicted by TUTR (Shi et al., 2023) and MID (Gu et al., 2022) are unfeasible (e.g., crossing the walls), predictions guided with MoDs follow human flow and implicitly respect the structure of the environment over long horizons.

ignores speed, NeMo-map jointly models both speed and orientation in a continuous manner, offering a more expressive motion prior. A qualitative example for a 60 s prediction is shown in Fig. 6. Human motion prediction using NeMo-map generates realistic trajectories that follow the complex topology of the environment, while deep learning-based results are unfeasible by crossing the walls.

Table 4: Long-term human motion prediction results on the ATC dataset with a prediction horizon of 60 s. We report ADE/FDE (mean $\pm$ one standard deviation), where lower values indicate better performance.

| Metric | **Ours** | CLiFF-LHMP | STeF-LHMP | TUTR | MID |
|---|---|---|---|---|---|
| ADE $\downarrow$ | **5.08 $\pm$ 4.28** | 5.45 $\pm$ 4.54 | 5.88 $\pm$ 5.55 | 12.10 $\pm$ 8.20 | 21.20 $\pm$ 5.28 |
| FDE $\downarrow$ | **10.97 $\pm$ 9.92** | 11.45 $\pm$ 10.20 | 12.24 $\pm$ 12.16 | 27.26 $\pm$ 19.23 | 44.47 $\pm$ 10.91 |

## 4.5 ABLATION STUDY

In this section, we analyze alternative methods for temporal encoding and further include hyper-parameter ablations on the SWGMM component number $J$ and the spatial feature grid resolution $H \times W$ in Appendix E.

In our method, we use a SIREN network to process the temporal input. For comparison, we evaluate two alternative mappings of time $t$ into a temporal feature vector $\mathbf{f}_t(t)$:

- **Temporal grid.** A learnable grid $\mathbf{G}_t \in \mathbb{R}^{K \times C_t}$ that captures daily periodicity, where $K$ is the number of discretized time bins (set to 24). The grid feature corresponding to each time bin is concatenated with the spatial feature and passed through an MLP with hidden sizes $[128, 64]$ and ReLU activations.

- **Fourier features.** The time input $t$ is mapped into a periodic embedding using Fourier features Tancik et al. (2020); Mildenhall et al. (2020). For $F$ frequencies, we construct

$$\mathbf{f}_t(t) = \left[ \sin(2^n 2\pi t), \ \cos(2^n 2\pi t) \right]_{n=0}^{F-1}.$$

  This representation enables the model to capture time-dependent variations at multiple resolutions. The implementation is identical to the temporal grid variant, except the time grid is replaced by Fourier features with $F = 4$, yielding an 8-dimensional temporal embedding.

Table 5 summarizes the results of the ablation study on temporal encoding. Replacing SIREN with a temporal grid or Fourier features results in higher NLL, confirming the advantage of using SIREN for modeling continuous temporal dynamics. Among the alternatives, Fourier features outperform the temporal grid, but both remain less accurate than SIREN. The reductions in NLL relative to our method are $0.082$ for the temporal grid and $0.063$ for Fourier features, with 95% confidence intervals.

Table 5: Comparing alternative time encodings for NeMo-map, again using the ATC dataset and comparing average negative log-likelihood (NLL) where lower indicates better accuracy. We report mean $\pm$ standard deviation, together with the reduction in NLL relative to our method and the corresponding 95% confidence interval (CI). All models are trained using the Adam optimizer with learning rate $10^{-3}$ for 100 epochs.

| Method | NLL $\downarrow$ | NLL reduction (vs Ours) | 95% CI |
|---|---|---|---|
| **Ours** | **0.775 $\pm$ 2.052** | – | – |
| Temporal grid | 0.857 $\pm$ 2.113 | +0.082 | [0.081, 0.083] |
| Fourier features | 0.838 $\pm$ 2.105 | +0.063 | [0.062, 0.064] |

## 5 CONCLUSIONS

We introduced the first-of-its-kind *continuous spatio-temporal map of dynamics* representation NeMo-map, a novel formulation of MoDs using implicit neural representations. In contrast to prior discretized methods such as CLiFF-map and STeF-map, our approach parametrizes a continuous neural function, which outputs the parameters of a Semi-Wrapped Gaussian Mixture Model at arbitrary spatio-temporal coordinates. The model enables smooth generalization across space and time, and provides a compact representation that avoids storing per-cell distributions.

Through experiments on the large-scale ATC dataset and the outdoor ETH/UCY dataset, we demonstrated that NeMo-map achieves substantially higher accuracy (lower negative log-likelihood) than existing MoD baselines, while offering practical map building times suitable for large-scale datasets. We also demonstrate that the learned flow fields capture multimodality, temporal variations, and environment topology without requiring explicit maps, and achieve higher performance in the trajectory prediction downstream task. Ablation studies confirmed the advantage of using SIREN-based temporal encoding over discrete or Fourier alternatives.

In summary, the results highlight continuous MoDs as a practical and scalable tool for modeling human motion dynamics. By combining accuracy, efficiency, and flexibility, the representation offers a powerful prior for downstream tasks such as socially aware navigation, long-term motion prediction, and localization in dynamic environments. Despite its advantages, the proposed method has limitations in handling sharp spatial or temporal discontinuities, such as temporary barriers or sudden event-driven changes. Adapting to such changes requires additional observations, which is a limitation shared by existing MoD approaches. In future work, we plan to extend this formulation with online update mechanisms to adapt continuously to evolving crowd behaviors, and to better handle sharp, event-driven discontinuities in motion patterns, further bridging the gap toward long-term real-world deployment.

## 6 ETHICS STATEMENT

The dataset used for training are publicly available and fully anonymized, representing persons only as 2D positions without identifiers or visual data. Maps of dynamics further aggregate these trajectories into statistical motion patterns, so no personal information are retained.

## 7 REPRODUCIBILITY STATEMENT

For reproducibility, we provide the full training and evaluation code together with detailed instructions in the supplementary material. The package includes our main model as well as the variants used in the ablation study. In addition, we provide scripts for generating and visualizing maps of dynamics (MoDs).

## ACKNOWLEDGEMENTS

This work received funding from the European Union's Horizon 2020 research and innovation programme under grant agreement No 101070596 (euRobin), and was supported by the Wallenberg AI, Autonomous Systems and Software Program (WASP) funded by the Knut and Alice Wallenberg Foundation.

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

## A    THE USE OF LARGE LANGUAGE MODELS (LLMS)

Parts of this manuscript were edited with the assistance of LLMs to improve grammar and clarity. All scientific content was written and verified by the authors.

## B    DATASET

**ATC (Brščić et al., 2013)**    This dataset provides sufficient multi-day coverage for evaluation. ATC was collected in a shopping mall in Japan using multiple 3D range sensors, recording pedestrian trajectories between 9:00 and 21:00, over a total of 92 days. ATC covers a large indoor environment, with a total area covered of approximately $900\,\mathrm{m}^2$. Because of the large scale of the dataset, we use first four days in the dataset (2012 Oct 24, 2012 Oct 28, 2012 Oct 31, and 2012 Nov 04) for experiments. The data from October 24 is used for training, while the other three days are used for evaluation. The observation rate is downsampled from over $10\,\mathrm{Hz}$ to $1\,\mathrm{Hz}$. After downsampling, the training set contains 717,875 recorded motion samples, and the test set contains 5,114,478 samples. ATC dataset environment layout is shown in Fig. 7.

**ETH (Pellegrini et al., 2009)/UCY (Lerner et al., 2007)**    The ETH/UCY datasets contain pedestrian trajectories captured in outdoor environments such as university campuses and shopping areas. We evaluate using all four scenes: ETH and HOTEL from the ETH dataset, and UNIV and ZARA from the UCY dataset.

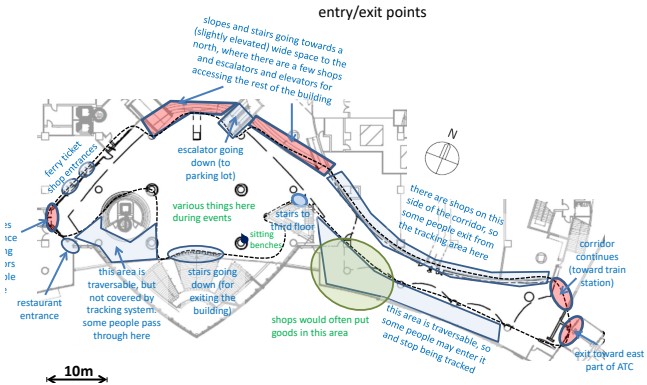

Figure 7: Layout of the ATC dataset environment (Brščić et al., 2013), showing the main east corridor and open areas annotated with semantic information such as entry and exit points, shops, seating areas, and stairs.

## C    IMPLEMENTATION DETAILS

The output of the network $\Phi_\theta$ parameterizes an SWGMM over speed and orientation. For $J$ mixture components, the network predicts $6J$ raw values per query coordinate. Each component $j$ is defined by: a mixture weight $w_j$, obtained by applying a softmax over the raw weights; a mean speed $\mu_{j,s} = \max(0, \tilde{\mu}_{j,s})$ and mean orientation $\mu_{j,a} = \tilde{\mu}_{j,a} \bmod 2\pi$; variances $\sigma^2_{j,s} = \exp(\mathrm{clamp}(\tilde{v}_{j,s}, -10, 10))$ and $\sigma^2_{j,a} = \exp(\mathrm{clamp}(\tilde{v}_{j,a}, -10, 10))$; and a correlation coefficient $\rho_j = 0.99\tanh(\tilde{\rho}_j)$. Altogether, the network defines a valid SWGMM with parameters as in Eq. (3), where $\boldsymbol{\mu}_j = (\mu_{j,s}, \mu_{j,a})$ and $\boldsymbol{\Sigma}_j$ is the covariance matrix with diagonal entries $\sigma^2_{j,s}$, $\sigma^2_{j,a}$ and correlation $\rho_j$. In the experiments, $J$ is set to 3 and coordinates are normalized to $[-1, 1]$. Spatial input is processed by an MLP with hidden sizes $[128, 64]$ and ReLU activations. In the ATC dataset, the environment exhibits strong daily regularity and the timestamps are provided in epoch format. We therefore convert them to time-of-day to capture long-term temporal patterns. In the ETH/UCY dataset where no such periodic structure is present, we instead use the frame as the temporal variable. Temporal input is processed by a two-layer SIREN (sine activations with $\omega_0^{(1)} = 30$ in the first

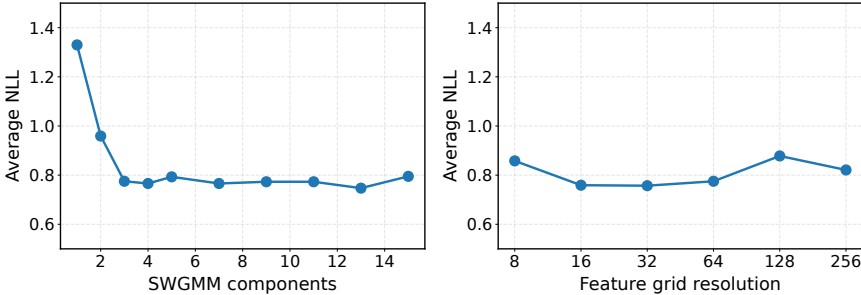

Figure 8: Hyperparameter ablations on the SWGMM component number $J$ and the spatial feature grid resolution $H \times W$.

layer and $\omega_0^{(h)} = 1$ in the hidden layer). The two streams are fused via FiLM modulation (Perez et al., 2018). The fused representation is passed to a linear head producing $6J$ outputs. Models are trained using the Adam optimizer with learning rate $10^{-3}$ for 100 epochs. An ablation of alternative temporal encodings is provided in Sec. 4.5.

## D    DOWNSTREAM TASK VALIDATION

In the experiments of long-term human motion prediction, for observations, we use $3\,\text{s}$ of each trajectory and use the remaining (up to the maximum prediction horizon) as the prediction ground truth. We compare our method against MoD-based predictors, Trajectory Unified TRansformer (TUTR, Shi et al. (2023)) and Motion Indeterminacy Diffusion (MID, Gu et al. (2022)).

TUTR unifies social interaction modeling and multimodal trajectory prediction components in a transformer encoder-decoder architecture. It predicts multiple trajectories with corresponding probabilities, and we evaluate using the trajectory with the highest probability. MID formulates trajectory prediction as a reverse process of motion indeterminacy diffusion, which gradually discards the indeterminacy to obtain desired trajectory from ambiguous walkable areas. Both MID and TUTR are trained for 100 epochs on both datasets.

For the evaluation metrics, we use *Average* and *Final Displacement Errors* (ADE and FDE). ADE describes the mean $L^2$ distance between predicted trajectories and the ground truth. FDE describes the $L^2$ distance between the predicted and the ground truth positions at the last prediction time step. For each ground truth trajectory we generate $k = 5$ prediction trajectories. Based on practical applications for autonomous robots, we evaluate using the most likely predicted trajectory. When probability information for the predicted trajectories is not available, such as in the case of the baseline MID, we report the mean ADE and FDE values over the predicted trajectories.

## E    HYPERPARAMETER ABLATION STUDY

For hyperparameter ablations, we study the effect of the SWGMM component number $J$ and the spatial feature grid resolution $H \times W$. Increasing $J$ allows the model to represent more complex human motion behaviors. However, a larger number of components also increases the output dimension of the MoD representation and leads to higher computational cost during both training and inference. In the left figure of Fig. 8, performance stabilizes when $J \geq 3$, showing that it is sufficient to capture the dominant multimodal motion patterns in human environments. Therefore, we recommend starting with $J = 3$ in practical applications.

For the spatial feature grid resolution $H \times W$, a finer grid enables better representation of local motion variations, but also increases memory consumption and training time. As shown in the right figure of Fig. 8, the performance remains stable in the range from $16 \times 16$ to $64 \times 64$. We therefore recommend starting in this range to have a good trade-off between performance and efficiency.

