# OpenReview forum: "NeMo-map: Neural Implicit Flow Fields for Spatio-Temporal Motion Mapping"
_ICLR.cc/2026/Conference — ICLR 2026 Poster_

### Official Review · Reviewer_ETtv · 2025-10-16

**Soundness:** 3
**Presentation:** 3
**Contribution:** 3
**Rating:** 8
**Confidence:** 2

**Summary:**

This paper proposes NeMo-map, a continuous neural implicit representation for modeling spatio-temporal motion dynamics in human environments. Unlike prior discretized Maps of Dynamics (MoDs) such as CLiFF-map and STeF-map, NeMo-map directly maps spatio-temporal coordinates to the parameters of a Semi-Wrapped Gaussian Mixture Model (SWGMM), enabling smooth, multimodal velocity distributions without grid discretization. The model combines a learnable spatial feature grid with a SIREN-based temporal encoder that captures daily periodic variations in human motion, producing continuous flow fields across both space and time. Experiments on a large-scale pedestrian tracking dataset show that NeMo-map achieves higher accuracy and significantly faster training than baseline MoD approaches, while qualitatively generating smooth, topologically consistent flow fields aligned with real-world environment structures. The approach is presented as a scalable and efficient foundation for dynamic scene understanding, motion prediction, and socially aware navigation.

**Strengths:**

The paper makes a clear and technically meaningful contribution by replacing discrete MoD representations with a continuous implicit field, eliminating the need for grid-based interpolation and yielding smooth, queryable flow estimates across both space and time. The probabilistic backbone—the Semi-Wrapped Gaussian Mixture Model—retains interpretability while enabling multimodal velocity representation, capturing correlations between speed and direction (Eq. 1–2). The use of SIREN-based temporal encoding is a strong design choice, effectively modeling daily periodicities and outperforming Fourier and discrete temporal grids in ablations (Table 3). Experimental validation is solid: the authors benchmark against multiple strong baselines (CLiFF-map, Online CLiFF, and STeF-map) and demonstrate both quantitative accuracy gains and significant efficiency improvements in training time (Table 2). The qualitative results are well-presented (Fig. 5), showing realistic spatio-temporal evolution of human flow patterns that align with semantic structure (e.g., benches, exits, corridors). Overall, the paper is technically rigorous, well-motivated, and carefully evaluated, making a strong contribution to spatio-temporal modeling in robotics.

**Weaknesses:**

1. The proposed implicit mapping formulation assumes a globally smooth function Φθ(x, t), which may be overly restrictive for environments with sharp spatial or temporal discontinuities (e.g., doorways, temporary barriers, or sudden event-driven flow changes). The model’s reliance on a fixed neural parameterization prevents local adaptation to abrupt changes, and the paper does not provide any mechanism (e.g., hierarchical grids or local residuals) to handle non-smooth motion transitions (Sec. 3.2).

2. The periodicity assumption in time modeling (Sec. 3.2, lines 215–218) simplifies human motion to a daily cycle, which is unrealistic for many public spaces with irregular or event-driven patterns. The evaluation dataset (ATC) naturally exhibits strong daily regularity, but no experiment tests NeMo-map on non-periodic or transient patterns, making the claimed generality questionable. The model’s performance could degrade if queried outside the trained temporal domain or under distribution shifts in time.

3. The evaluation metric focuses solely on negative log-likelihood (Sec. 4.4, Table 1), which measures statistical fit but not flow-level accuracy or spatial consistency—key factors for downstream robotic use. The paper reports no task-based or trajectory-level validation (e.g., motion prediction or navigation performance), making it difficult to assess whether the smoother velocity fields actually improve practical planning or prediction outcomes.

4. While the authors emphasize NeMo-map’s faster training (Table 2), the comparison fairness is limited: CLiFF-map is retrained per hour and cell, while NeMo-map jointly trains across space and time with GPU acceleration. The paper does not normalize for total computation per prediction or per environment, so the reported “two orders of magnitude speedup” may overstate the practical advantage. Moreover, the ablation study (Sec. 4.6) isolates temporal encodings but omits critical architecture factors such as mixture component count (J) or grid resolution (Gs), leaving the method’s scalability characteristics underexplored.

**Questions:**

please address the concerns above

---

> ### Author Response · Authors · 2025-11-24
>
> Thank you for your valuable feedback to help us improve the work. We have updated the manuscript accordingly. The detailed responses to the questions and concerns are provided below:
>
> **Q1: Periodicity assumption & lack of non-periodic validation**
> >The periodicity assumption in time modeling (Sec. 3.2, lines 215–218) simplifies human motion to a daily cycle, which is unrealistic for many public spaces with irregular or event-driven patterns. The evaluation dataset (ATC) naturally exhibits strong daily regularity, but no experiment tests NeMo-map on non-periodic or transient patterns, making the claimed generality questionable. The model’s performance could degrade if queried outside the trained temporal domain or under distribution shifts in time.
>
> A1: We thank the reviewer for this insightful comment. We agree that evaluating only on ATC introduces a strong daily periodicity assumption on the temporal input. In the revised version, we relax this assumption and additionally evaluate our method on the ETH/UCY dataset, which contains non-periodic motion patterns in outdoor environments. This demonstrates that NeMo-map can also operate effectively without daily periodic structure.
>
> As shown in Table 1 below, NeMo-map achieves lower NLL across scenes. All improvements are statistically significant ($p<0.001$, paired t-tests).
>
> **Table 1: NLL$\downarrow$ on ETH/UCY dataset (mean ± std.)**
>
> | Method | ETH | HOTEL | UNIV | ZARA |
> |--------|-----|-------|------|------|
> | **NeMo-map (ours)** | **−0.384 ± 2.051** | **−0.838 ± 4.043** | **0.404 ± 1.902** | **−0.342 ± 2.152** |
> | CLiFF-map | 0.112 ± 4.005 | 0.701 ± 4.533 | 0.518 ± 2.125 | 0.068 ± 4.265 |
> | Online CLiFF-map | 0.086 ± 4.451 | 1.241 ± 7.142 | 0.577 ± 2.548 | 0.186 ± 5.477 |
> | STeF-map | 2.315 ± 6.016 | 3.349 ± 7.660 | 10.932 ± 12.771 | 2.784 ± 7.014 |
>
> In the revised paper, we have updated the Sec. 4.2 and Sec. 4.3 to include quantitative and qualitative results of the ETH/UCY datasets.
>
>
> **Q2: Lack of downstream task evaluation**
> >The evaluation metric focuses solely on negative log-likelihood (Sec. 4.4, Table 1), which measures statistical fit but not flow-level accuracy or spatial consistency—key factors for downstream robotic use. The paper reports no task-based or trajectory-level validation (e.g., motion prediction or navigation performance), making it difficult to assess whether the smoother velocity fields actually improve practical planning or prediction outcomes.
>
>
> A2: We thank the reviewer for this insightful comment. To evaluate whether our proposed approach benefits downstream task and to demonstrate the practical utility of our method, we have added additional downstream evaluations on the long-term human motion prediction (LHMP) task. We compare the performance of our method with prior MoD-based predictors, CLiFF-LHMP (Zhu et al., 2023), STeF-LHMP (Molina et al., 2022), and also compare with a transformer-based predictor (TUTR, Shi et al., 2023) and a diffusion-based predictor (MID, Gu et al., 2022).
>
> The prediction accuracy results on the ATC dataset with a prediction horizon of 60s are shown in Table 2. Evaluation metrics are *Average* and *Final Displacement Errors* (ADE and FDE), where lower values indicate better prediction accuracy. The improvements arise from NeMo-map's ability to capture smooth and precise local flow patterns, resulting in more accurate predictions. This new experiment has been added in Sec. 4.4. Results across multiple prediction horizons (10s–60s) and qualitative prediction examples are included in Fig. 6 of the revised paper.
>
>
> **Table 2: Long-term human motion prediction results with a prediction horizon of 60s. We report ADE/FDE (mean ± std.)**
> | Method | ADE$\downarrow$ | FDE$\downarrow$ |
> |--------|------|------|
> | **NeMo-map (ours)** | **5.08 ± 4.28** | **10.97 ± 9.92** |
> | CLiFF-LHMP | 5.45 ± 4.54 | 11.45 ± 10.20 |
> | STeF-LHMP | 5.88 ± 5.55 | 12.24 ± 12.16 |
> | TUTR | 12.10 ± 8.20 | 27.26 ± 19.23 |
> | MID | 21.20 ± 5.28 | 44.47 ± 10.91 |

---

> ### Author Response · Authors · 2025-11-24
>
> **Q3: Training-time comparison**
> >While the authors emphasize NeMo-map’s faster training (Table 2), the comparison fairness is limited: CLiFF-map is retrained per hour and cell, while NeMo-map jointly trains across space and time with GPU acceleration. The paper does not normalize for total computation per prediction or per environment, so the reported “two orders of magnitude speedup” may overstate the practical advantage.
>
> A3: We thank the reviewer for this insightful comment. We agree that the training time comparison was not normalized due to the differences in how two methods operate. CLiFF-map requires fitting a separate model for every spatial grid cell and every time slice, which leads to substantial cumulative training cost. NeMo-map trains a model over spatio-temporal domain and supports inference at arbitrary locations. To avoid overstating, we have revised training time claims in the paper in Sec. 1, Sec. 4.2, and Sec. 5.
>
> **Q4: Missing ablations**
> >Moreover, the ablation study (Sec. 4.6) isolates temporal encodings but omits critical architecture factors such as mixture component count (J) or grid resolution (Gs), leaving the method’s scalability characteristics underexplored.
>
> A4: We thank the reviewer for this insightful comment. In the revised version, we have extended the ablation study to include the hyperparameters of SWGMM mixture components $J$ and the spatial feature grid resolution $H\times W$. The updated results are in Sec. 4.5 and detailed in Appendix E.
>
> The experiments show that 1) increasing $J$ improves representation accuracy initially, and performance stabilizes once $J\geq3$, showing that starting from $J=3$ is an effective default setting in practice. 2) For the spatial feature grid resolution $H \times W$, a finer grid enables better representation of local motion variations, but also increases memory consumption and training time. The performance remains stable in the range from $16\times16$ to $64\times64$. We therefore recommend starting in this range to have a good trade-off between performance and efficiency.
>
>
> **Q5: Difficulty with discontinuities**
> >The proposed implicit mapping formulation assumes a globally smooth function Φθ(x, t), which may be overly restrictive for environments with sharp spatial or temporal discontinuities (e.g., doorways, temporary barriers, or sudden event-driven flow changes). The model’s reliance on a fixed neural parameterization prevents local adaptation to abrupt changes, and the paper does not provide any mechanism (e.g., hierarchical grids or local residuals) to handle non-smooth motion transitions (Sec. 3.2).
>
> A5: We thank the reviewer for this insightful suggestion. We agree that the proposed method face limitations when modeling sharp spatial or temporal discontinuities, such as temporary barriers or event-driven changes. In such cases, additional observations reflecting the new dynamics would be required for the model to adapt. This limitation also affects existing MoD-based baselines. We have added this discussion to the limitations and future work in Sec. 5.

---

### Official Review · Reviewer_xjtj · 2025-10-23

**Soundness:** 2
**Presentation:** 3
**Contribution:** 2
**Rating:** 4
**Confidence:** 3

**Summary:**

This paper proposes a new way to represent how humans move in an environment. It introduces a continuous map of dynamics called NeMo-map, which models motion as a smooth function across space and time instead of using fixed grids. The method uses an implicit neural network to predict local motion distributions based on spatial and temporal inputs. This continuous approach preserves detailed motion patterns, enables flexible querying at any point in time or location, and avoids the data loss caused by discretization

**Strengths:**

- The proposed NeMo-map, unlike existing methods such as CLiFF-map and STeF-map, continuously models space and time, enabling smooth and generalizable representations of human mobility patterns.
- By leveraging implicit neural representation, it achieves both high expressive power and computational efficiency, while reducing map construction time by more than an order of magnitude compared to previous approaches.

**Weaknesses:**

- From the perspective of generalization, there remains considerable room to assess the significance of the proposed methodology.
The study was validated only on the ATC dataset. Although, as shown in Figure 4, it includes various indoor scenarios and conditions, it still represents a single dataset distribution. Therefore, additional experiments and performance evaluations on other datasets are necessary.

- Since the proposed NeMo-map heavily relies on neural implicit representation, its internal representations are limited in terms of interpretability. Consequently, it is challenging to analyze the model’s decision-making process or evaluate its reliability in real-world robotic operation environments.

**Questions:**

- In Line 146, the authors explicitly state that the proposed method is fundamentally different from trajectory prediction models. However, in Figure 1, to illustrate the application of MoD, the authors present results alongside existing human trajectory prediction models such as Social LSTM and MID. This raises the question of whether such comparisons should be included as part of actual experimental validation, rather than being presented merely as conceptual illustrations. Furthermore, as mentioned in Line 036, “a planner informed by MoDs can exploit prior knowledge of human motion patterns to generate a trajectory that aligns with the expected flow, allowing the robot to reach the goal safely and efficiently. MoDs can also be applied to long-term human motion prediction (Zhu et al., 2023). As shown in the right of Fig. 1, MoDs help predict realistic trajectories that implicitly respect the complex topology of the environment, such as navigating around corners or avoiding obstacles.” Given this description, it would be valuable to verify whether, in the field of human motion prediction or human trajectory prediction, the proposed approach indeed provides better priors that contribute to the performance improvements observed in “With MoD guidance” in Figure 1. Such experiments would substantially reinforce the credibility of the claim that MoD guidance effectively aids motion planning and human motion prediction.

- The authors conducted experiments on the ATC dataset and included the analysis in Section 4.5 (Qualitative Results). However, the explanation could be more detailed. Figure 5 merely shows the dataset distribution across different time periods, but it would be insightful to elaborate on what specific human lifestyle patterns lead to certain behaviors or activities at particular locations and times. Moreover, it would be helpful to clarify under which scenarios(situations) or environmental conditions the model’s predictions are accurate and where they fail. A deeper discussion on these aspects would strengthen the interpretability and practical relevance of the results.

---

> ### Author Response · Authors · 2025-11-24
>
> Thank you for your valuable feedback to help us improve the work. We have updated the manuscript accordingly. The detailed responses to the questions and concerns are provided below:
>
> **Q1: Limited dataset diversity**
> >From the perspective of generalization, there remains considerable room to assess the significance of the proposed methodology. The study was validated only on the ATC dataset. Although, as shown in Figure 4, it includes various indoor scenarios and conditions, it still represents a single dataset distribution. Therefore, additional experiments and performance evaluations on other datasets are necessary.
>
>
> A1: We thank the reviewer for this insightful comment. Following the comment, we conducted experiments on the additional ETH/UCY dataset, which contains trajectories captured in outdoor environments.
>
> As shown in Table 1 below, NeMo-map achieves lower NLL across scenes. All improvements are statistically significant ($p<0.001$, paired t-tests).
>
>
> **Table 1: NLL$\downarrow$ on ETH/UCY dataset (mean ± std.)**
>
> | Method | ETH | HOTEL | UNIV | ZARA |
> |--------|-----|-------|------|------|
> | **NeMo-map (ours)** | **−0.384 ± 2.051** | **−0.838 ± 4.043** | **0.404 ± 1.902** | **−0.342 ± 2.152** |
> | CLiFF-map | 0.112 ± 4.005 | 0.701 ± 4.533 | 0.518 ± 2.125 | 0.068 ± 4.265 |
> | Online CLiFF-map | 0.086 ± 4.451 | 1.241 ± 7.142 | 0.577 ± 2.548 | 0.186 ± 5.477 |
> | STeF-map | 2.315 ± 6.016 | 3.349 ± 7.660 | 10.932 ± 12.771 | 2.784 ± 7.014 |
>
> In the revised paper, we have updated the Sec. 4.2 and Sec. 4.3 to include quantitative and qualitative results of the ETH/UCY datasets.
>
>
> **Q2: Downstream task evaluation**
> >Since the proposed NeMo-map heavily relies on neural implicit representation, its internal representations are limited in terms of interpretability. Consequently, it is challenging to analyze the model’s decision-making process or evaluate its reliability in real-world robotic operation environments. In Line 146, the authors explicitly state that the proposed method is fundamentally different from trajectory prediction models. However, in Figure 1, to illustrate the application of MoD, the authors present results alongside existing human trajectory prediction models such as Social LSTM and MID. This raises the question of whether such comparisons should be included as part of actual experimental validation, rather than being presented merely as conceptual illustrations. Furthermore, as mentioned in Line 036, “a planner informed by MoDs can exploit prior knowledge of human motion patterns to generate a trajectory that aligns with the expected flow, allowing the robot to reach the goal safely and efficiently. MoDs can also be applied to long-term human motion prediction (Zhu et al., 2023). As shown in the right of Fig. 1, MoDs help predict realistic trajectories that implicitly respect the complex topology of the environment, such as navigating around corners or avoiding obstacles.” Given this description, it would be valuable to verify whether, in the field of human motion prediction or human trajectory prediction, the proposed approach indeed provides better priors that contribute to the performance improvements observed in “With MoD guidance” in Figure 1. Such experiments would substantially reinforce the credibility of the claim that MoD guidance effectively aids motion planning and human motion prediction.
>
> A2: We thank the reviewer for this insightful comment.
>
> To address whether our proposed approach provides better priors for downstream task and to demonstrate the practical utility of our method, we have added additional downstream evaluations on the long-term human motion prediction (LHMP) task. We compare the performance of our method with prior MoD-based predictors, CLiFF-LHMP (Zhu et al., 2023), STeF-LHMP (Molina et al., 2022), and also compare with a transformer-based predictor (TUTR, Shi et al., 2023) and a diffusion-based predictor (MID, Gu et al., 2022).
>
> The prediction accuracy results on the ATC dataset with a prediction horizon of 60s are shown in Table 2. Evaluation metrics are *Average* and *Final Displacement Errors* (ADE and FDE), where lower values indicate better prediction accuracy. The improvements arise from NeMo-map's ability to capture smooth and precise local flow patterns, resulting in more accurate predictions. This new experiment has been added in Sec. 4.4. Results across multiple prediction horizons (10s–60s) and qualitative prediction examples are included in Fig. 6 of the revised paper.
>
>
> **Table 2: Long-term human motion prediction results with a prediction horizon of 60s. We report ADE/FDE (mean ± std.)**
> | Method | ADE$\downarrow$ | FDE$\downarrow$ |
> |--------|------|------|
> | **NeMo-map (ours)** | **5.08 ± 4.28** | **10.97 ± 9.92** |
> | CLiFF-LHMP | 5.45 ± 4.54 | 11.45 ± 10.20 |
> | STeF-LHMP | 5.88 ± 5.55 | 12.24 ± 12.16 |
> | TUTR | 12.10 ± 8.20 | 27.26 ± 19.23 |
> | MID | 21.20 ± 5.28 | 44.47 ± 10.91 |

---

> > ### Author Response · Authors · 2025-11-24
> >
> > **Q3: Qualitative explanation and interpretability**
> > >The authors conducted experiments on the ATC dataset and included the analysis in Section 4.5 (Qualitative Results). However, the explanation could be more detailed. Figure 5 merely shows the dataset distribution across different time periods, but it would be insightful to elaborate on what specific human lifestyle patterns lead to certain behaviors or activities at particular locations and times. Moreover, it would be helpful to clarify under which scenarios(situations) or environmental conditions the model’s predictions are accurate and where they fail. A deeper discussion on these aspects would strengthen the interpretability and practical relevance of the results.
> >
> > A3: We thank the reviewer for this insightful suggestion. Following the comment, we have updated the qualitative analysis in Sec. 4.3 of the revised paper. The updated section includes multiple representative human motion scenarios and provides detailed comparisons showing how our method models these motion patterns relative to baseline MoDs.
> >
> > These comparisons show that NeMo-map:
> >  - maintains continuous and coherent flow fields in crossing traffic structures, models multimodal motion patterns consistently (Fig. 3)
> >  - adapts to temporal variations in pedestrian motion patterns throughout the day (Fig. 4)
> >  - captures emerging directional trends and captures diverse motion patterns in multiple environments (Fig. 5)

---

### Official Review · Reviewer_g9DM · 2025-11-01

**Soundness:** 3
**Presentation:** 3
**Contribution:** 3
**Rating:** 6
**Confidence:** 2

**Summary:**

This paper introduces a novel approach called Neural Motion Map (NeMo-map) for modeling human motion patterns in complex environments. Unlike traditional Maps of Dynamics (MoDs) that rely on discrete spatial grids, NeMo-map uses a continuous spatio-temporal implicit neural representation. It maps any given (x, y, t) coordinate to the parameters of a Semi-Wrapped Gaussian Mixture Model (SWGMM), enabling smooth and multimodal modeling of velocity distributions. The method is evaluated on the large-scale ATC pedestrian dataset, demonstrating superior accuracy, smoother flow fields, and significantly faster map construction compared to existing baselines such as CLiFF-map and STeF-map.

**Strengths:**

- This work to apply implicit neural representations to the MoD problem, offering a continuous and differentiable alternative to grid-based methods.
- The use of SWGMM allows joint modeling of speed and orientation, capturing multimodal and correlated motion patterns effectively.
- The method achieves the lowest NLL on the ATC dataset and is orders of magnitude faster in training than CLiFF-map.

**Weaknesses:**

- While the model implicitly learns spatial constraints, it does not explicitly incorporate geometric or semantic map information, which could improve robustness in complex environments.
- The model assumes daily periodicity in motion patterns, which may not hold in environments with irregular or event-driven dynamics.
- The proposed method is trained offline and does not support incremental updates or online learning, limiting its applicability in non-stationary environments

**Questions:**

This paper presents an effective approach to modeling human motion dynamics using continuous neural representations. The method significantly outperforms existing baselines in both accuracy and efficiency. While there are areas for improvement—particularly in online adaptability, structural integration. Please see the weakness.

And I would like to clarify that I am not a specialist in this particular field. Therefore, my review reflects my understanding and interpretation as a general reviewer, and I hope my comments are still helpful.

---

> ### Author Response · Authors · 2025-11-24
>
> Thank you for your valuable feedback to help us improve the work. We have updated the manuscript accordingly. The detailed responses to the questions and concerns are provided below:
>
> **Q1: Daily periodicity**
> >The model assumes daily periodicity in motion patterns, which may not hold in environments with irregular or event-driven dynamics.
>
> A1: We thank the reviewer for this insightful comment. Following the comment, we conducted additional experiments. We agree that evaluating only on ATC introduces a strong daily periodicity assumption on the temporal input. In the revised version, we relax this assumption and additionally evaluate our method on the ETH/UCY dataset, which contains non-periodic motion patterns in outdoor environments. This demonstrates that NeMo-map can also operate effectively without daily periodic structure.
>
>
> As shown in Table 1 below, NeMo-map achieves lower NLL across scenes. All improvements are statistically significant ($p<0.001$, paired t-tests).
>
>
> **Table 1: NLL$\downarrow$ on ETH/UCY dataset (mean ± std.)**
>
> | Method | ETH | HOTEL | UNIV | ZARA |
> |--------|-----|-------|------|------|
> | **NeMo-map (ours)** | **−0.384 ± 2.051** | **−0.838 ± 4.043** | **0.404 ± 1.902** | **−0.342 ± 2.152** |
> | CLiFF-map | 0.112 ± 4.005 | 0.701 ± 4.533 | 0.518 ± 2.125 | 0.068 ± 4.265 |
> | Online CLiFF-map | 0.086 ± 4.451 | 1.241 ± 7.142 | 0.577 ± 2.548 | 0.186 ± 5.477 |
> | STeF-map | 2.315 ± 6.016 | 3.349 ± 7.660 | 10.932 ± 12.771 | 2.784 ± 7.014 |
>
> In the revised paper, we have updated the Sec. 4.2 and Sec. 4.3 to include quantitative and qualitative results of the ETH/UCY datasets.
>
>
> **Q2: Geometric/semantic map input**
> >While the model implicitly learns spatial constraints, it does not explicitly incorporate geometric or semantic map information, which could improve robustness in complex environments.
>
>
> A2: We thank the reviewer for this insightful comment. It is true that the current model does not explicitly incorporate geometric or semantic map information. In the MoD literature, two modeling directions are commonly explored:
>  - Trajectory-only MoDs (like ours), which learn motion patterns directly from observed human trajectories and require no map annotations.
>  - Map-only MoDs, which infer plausible motion patterns purely from geometric maps without any trajectory data.
>
> We agree that incorporating geometric or semantic maps as additional information could improve performance in complex or highly structured environments. However, this comes at the cost of increased input requirements, limiting its applicability.
>
>
> **Q3: Online update**
> >The proposed method is trained offline and does not support incremental updates or online learning, limiting its applicability in non-stationary environments
>
>
> A3: We thank the reviewer for this insightful comment. We agree that the current method has limitations in handling sharp spatial or temporal discontinuities, such as temporary barriers or sudden event-driven changes, in non-stationary environments. Extending the framework with online or incremental update mechanisms would make the approach more practical and closer to long-term real-world deployment. We have added this limitation and the corresponding future work discussion in the revised paper.

---

### Official Review · Reviewer_9dui · 2025-11-01

**Soundness:** 3
**Presentation:** 2
**Contribution:** 2
**Rating:** 4
**Confidence:** 4

**Summary:**

This paper proposes a continuous representation for Maps of Dynamics (MoDs) that models statistical human motion patterns using implicit neural functions rather than traditional discrete spatial grids.

The authors replace grid-based discretization with a neural implicit function that maps continuous spatio-temporal coordinates ((x, t)) directly to the parameters of a Semi-Wrapped Gaussian Mixture Model (SWGMM), capturing both linear (speed) and circular (orientation) components of human motion. This approach allows smooth generalization across space and time and removes the need for manual resolution tuning or interpolation in sparsely sampled regions.

The model architecture combines:

* A learnable spatial feature grid queried via bilinear interpolation.
* A temporal encoder using SIREN (sinusoidal representation networks) to model daily periodicity.
* A fully connected MLP that outputs SWGMM parameters for multimodal, continuous motion distributions.

Experiments on the ATC pedestrian dataset* demonstrate that NeMo-map achieves:

* The lowest negative log-likelihood (NLL = 0.775 ± 2.052), outperforming established baselines such as CLiFF-map, Online CLiFF-map, and STeF-map.
* Smooth temporal adaptation of motion fields without explicit geometric maps.

In summary, the paper’s key contribution is the NeMo-map, the first continuous spatio-temporal MoD representation using implicit neural fields.

**Strengths:**

The paper addresses the limitation of existing Maps of Dynamics (MoDs)—their reliance on discrete spatial grids—by introducing a continuous implicit neural representation. While this idea of applying neural implicit fields to motion modeling is an incremental extension of trends in continuous scene representations (e.g., NeRF, SIREN), it is relatively new within the specific context of MoD construction for human motion modeling.

The technical formulation is mathematically sound and builds on established probabilistic foundations (Semi-Wrapped Gaussian Mixture Models). The experimental setup uses a large-scale and reputable dataset (ATC) and provides clear quantitative comparisons with several reasonable baselines (CLiFF-map, Online CLiFF-map, and STeF-map).

The paper is generally well-written and structured, with clear motivations, illustrative figures (e.g., flow-field visualizations in Fig. 5). The proposed NeMo-map provides a modest but meaningful step forward for motion-aware mapping, particularly for robotics tasks requiring real-time adaptation to human flow patterns. Its computational efficiency and smoothness could make it a practical alternative to existing grid-based approaches in large or dynamic environments.

**Weaknesses:**

1. Limited novelty and conceptual contribution: While the paper presents a “continuous” MoD via implicit neural functions, the underlying methodological idea—using an implicit neural field to map coordinates to local probabilistic parameters—is conceptually similar to existing implicit representation frameworks (e.g., NeRF, SIREN, and Neural Fields for flow estimation). The novelty primarily lies in adapting these tools to an existing MoD formulation rather than introducing fundamentally new modeling principles. In this sense, NeMo-map reads as an engineering refinement of the CLiFF-map rather than a conceptual breakthrough.

2. Narrow experimental validation and lack of domain diversity: The evaluation is conducted entirely on a single dataset (ATC), which—while large—is limited to indoor pedestrian motion in a controlled shopping mall environment. The paper’s claims about generalization in spatio-temporal motion fields would be more convincing if validated across domains (e.g., outdoor crowd datasets, vehicle or mixed-agent motion). Testing only on ATC raises concerns that the model’s performance gains might be dataset-specific, particularly since temporal periodicity (daily cycles) is a strong prior in ATC but not in all motion environments. Expanding experiments to additional datasets (e.g., ETH/UCY or MOTChallenge) would considerably strengthen the generality claim.

3. Lack of downstream evaluation or task relevance: Although the paper emphasizes applications such as socially aware navigation and long-term prediction, the evaluation is limited to negative log-likelihood (NLL) metrics. These metrics assess representational fit but not utility. It remains unclear whether the proposed NeMo-map actually improves downstream performance in motion planning, collision avoidance, or human trajectory forecasting compared with prior MoD-based methods. Including an evaluation in a planning or prediction pipeline—e.g., comparing robot navigation success rates or forecast accuracy using NeMo vs. CLiFF—would provide stronger evidence of practical benefit.

4. Inadequate comparative visualization of results: The qualitative results (Fig. 5) present visualizations only for the proposed NeMo-map, without showing side-by-side comparisons with baseline methods such as CLiFF-map, STeF-map, or Online CLiFF-map. This omission makes it difficult for readers to visually assess the claimed advantages in smoothness, continuity, or multimodality. Including qualitative comparisons—e.g., overlaying the predicted flow fields of different models in the same regions—would provide a much clearer and more compelling demonstration of NeMo-map’s improvements.

**Questions:**

1. Could the authors provide qualitative visual comparisons between NeMo-map and the baselines (e.g., CLiFF-map, Online CLiFF-map, STeF-map)?
2. How well does NeMo-map generalize to motion domains with very different spatial and temporal dynamics (e.g., outdoor environments, vehicle flows, or multi-agent interactions)?
3. Since one of the motivations is to support robot navigation and trajectory prediction, could the authors show how NeMo-map impacts these tasks compared to prior MoD-based representations?

---

> ### Author Response · Authors · 2025-11-24
>
> Thank you for your valuable feedback to help us improve the work. We have updated the manuscript accordingly. The detailed responses to the questions and concerns are provided below:
>
> **Q1: Narrow experimental validation and lack of domain diversity**
> >How well does NeMo-map generalize to motion domains with very different spatial and temporal dynamics (e.g., outdoor environments, vehicle flows, or multi-agent interactions)? The evaluation is conducted entirely on a single dataset (ATC), which—while large—is limited to indoor pedestrian motion in a controlled shopping mall environment. The paper’s claims about generalization in spatio-temporal motion fields would be more convincing if validated across domains (e.g., outdoor crowd datasets, vehicle or mixed-agent motion). Testing only on ATC raises concerns that the model’s performance gains might be dataset-specific, particularly since temporal periodicity (daily cycles) is a strong prior in ATC but not in all motion environments. Expanding experiments to additional datasets (e.g., ETH/UCY or MOTChallenge) would considerably strengthen the generality claim.
>
> A1:
> We thank the reviewer for this insightful comment. To demonstrate that NeMo-map generalizes beyond indoor environments (ATC shopping mall) and does not rely on daily periodicity priors, we conducted additional experiments on the suggested ETH/UCY dataset, which contains trajectories captured in outdoor environments.
>
> As shown in Table 1 below, NeMo-map achieves lower NLL across scenes. All improvements are statistically significant ($p<0.001$, paired t-tests).
>
> **Table 1: NLL$\downarrow$ on ETH/UCY dataset (mean ± std.)**
>
> | Method | ETH | HOTEL | UNIV | ZARA |
> |--------|-----|-------|------|------|
> | **NeMo-map (ours)** | **−0.384 ± 2.051** | **−0.838 ± 4.043** | **0.404 ± 1.902** | **−0.342 ± 2.152** |
> | CLiFF-map | 0.112 ± 4.005 | 0.701 ± 4.533 | 0.518 ± 2.125 | 0.068 ± 4.265 |
> | Online CLiFF-map | 0.086 ± 4.451 | 1.241 ± 7.142 | 0.577 ± 2.548 | 0.186 ± 5.477 |
> | STeF-map | 2.315 ± 6.016 | 3.349 ± 7.660 | 10.932 ± 12.771 | 2.784 ± 7.014 |
>
> In the revised paper, we have updated the Sec. 4.2 and Sec. 4.3 to include quantitative and qualitative results of the ETH/UCY datasets.
>
> **Q2: Lack of downstream task evaluation**
> >Since one of the motivations is to support robot navigation and trajectory prediction, could the authors show how NeMo-map impacts these tasks compared to prior MoD-based representations? Although the paper emphasizes applications such as socially aware navigation and long-term prediction, the evaluation is limited to negative log-likelihood (NLL) metrics. These metrics assess representational fit but not utility. It remains unclear whether the proposed NeMo-map actually improves downstream performance in motion planning, collision avoidance, or human trajectory forecasting compared with prior MoD-based methods. Including an evaluation in a planning or prediction pipeline—e.g., comparing robot navigation success rates or forecast accuracy using NeMo vs. CLiFF—would provide stronger evidence of practical benefit.
>
> A2: We thank the reviewer for this insightful comment.
>
> To demonstrate the practical utility of our method, we conducted additional downstream evaluations on the long-term human motion prediction (LHMP) task. We compare the performance of our method with prior MoD-based predictors, CLiFF-LHMP (Zhu et al., 2023), STeF-LHMP (Molina et al., 2022), and also compare with a transformer-based predictor (TUTR, Shi et al., 2023) and a diffusion-based predictor (MID, Gu et al., 2022).
>
> The prediction accuracy results on the ATC dataset with a prediction horizon of 60s are shown in Table 2. Evaluation metrics are *Average* and *Final Displacement Errors* (ADE and FDE), where lower values indicate better prediction accuracy. The improvements arise from NeMo-map's ability to capture smooth and precise local flow patterns, resulting in more accurate predictions. This new experiment has been added in Sec. 4.4. Results across multiple prediction horizons (10s–60s) and qualitative prediction examples are included in Fig. 6 of the revised paper.
>
>
> **Table 2: Long-term human motion prediction results with a prediction horizon of 60s. We report ADE/FDE (mean ± std.)**
> | Method | ADE$\downarrow$ | FDE$\downarrow$ |
> |--------|------|------|
> | **NeMo-map (ours)** | **5.08 ± 4.28** | **10.97 ± 9.92** |
> | CLiFF-LHMP | 5.45 ± 4.54 | 11.45 ± 10.20 |
> | STeF-LHMP | 5.88 ± 5.55 | 12.24 ± 12.16 |
> | TUTR | 12.10 ± 8.20 | 27.26 ± 19.23 |
> | MID | 21.20 ± 5.28 | 44.47 ± 10.91 |

---

> ### Author Response · Authors · 2025-11-24
>
> **Q3: Inadequate comparative visualization of results**
> >Could the authors provide qualitative visual comparisons between NeMo-map and the baselines (e.g., CLiFF-map, Online CLiFF-map, STeF-map)? The qualitative results (Fig. 5) present visualizations only for the proposed NeMo-map, without showing side-by-side comparisons with baseline methods such as CLiFF-map, STeF-map, or Online CLiFF-map. This omission makes it difficult for readers to visually assess the claimed advantages in smoothness, continuity, or multimodality. Including qualitative comparisons—e.g., overlaying the predicted flow fields of different models in the same regions—would provide a much clearer and more compelling demonstration of NeMo-map’s improvements.
>
> A3: We thank the reviewer for this helpful suggestion. Following the comment, we have updated the qualitative result in Sec. 4.3 in the revised paper, to include side-by-side visual comparisons between NeMo-map and all baseline methods on both the ATC and ETH/UCY datasets. NLL heatmaps are also provided to show the representation accuracy.
> These comparisons show that NeMo-map:
>  - maintains continuous and coherent flow fields in crossing traffic structures, models multimodal motion patterns consistently (Fig. 3)
>  - adapts to temporal variations in pedestrian motion patterns throughout the day (Fig. 4)
>  - captures emerging directional trends and captures diverse motion patterns in multiple environments (Fig. 5)

---

### Author Response · Authors · 2025-11-24
**General Response and Summary of Revisions**

We thank Area Chair for the time and effort, and we thank all reviewers for their valuable feedback that helped us further improve the paper. In the submission, we provided the full training and evaluation code for reproducibility. In the revision, we uploaded a revised manuscript, updated according to the reviewers' comments, with all changes highlighted in blue. Below, we summarize the **strengths** highlighted by reviewers, as well as the **concerns** raised and how we have addressed them.

---

### **Strengths**:
1.  **Soundness**:
       - "The technical formulation is mathematically sound. Provides clear quantitative comparisons with several reasonable baselines." `9dui`
     - "Experimental validation is solid: the authors benchmark against multiple strong baselines (CLiFF-map, Online CLiFF, and STeF-map)."  `ETtv`

2.  **Presentation**:
     - "The paper is generally well-written and structured, with clear motivations, illustrative figures." `9dui`
     - "The qualitative results are well-presented." `ETtv`

3.  **Contribution**:
       - "Overall, the paper is technically rigorous, well-motivated, and carefully evaluated, making a strong contribution to spatio-temporal modeling in robotics." `ETtv`

       - "The proposed NeMo-map provides a modest but meaningful step forward for motion-aware mapping, particularly for robotics tasks requiring real-time adaptation to human flow patterns." `9dui`

       - "By leveraging implicit neural representation, it achieves both high expressive power and computational efficiency." `xjtj`

       - "The method significantly outperforms existing baselines in both accuracy and efficiency." `g9DM`

---

### **Concerns and improvements**:
Reviewers' comments greatly helped us strengthen the validation of our method and improve the presentation. Based on the comments, we have strengthened the paper through additional experiments, improved visualizations, and clearer discussions to address the raised concerns in four main aspects:

1. **Broader validation and generalization** (Reviewers: `9dui`, `g9DM`, `xjtj` and `ETtv`)
    - Reviewer concerns:
        > `9dui`: How well does NeMo-map generalize to motion domains with very different spatial and temporal dynamics (e.g., outdoor environments, vehicle flows, or multi-agent interactions)? Testing only on ATC raises concerns that the model’s performance gains might be dataset-specific, particularly since temporal periodicity (daily cycles) is a strong prior in ATC but not in all motion environments. Expanding experiments to additional datasets (e.g., ETH/UCY or MOTChallenge) would considerably strengthen the generality claim.

        > `g9DM`: The model assumes daily periodicity in motion patterns, which may not hold in environments with irregular or event-driven dynamics.

        > `xjtj`: The study was validated only on the ATC dataset. It still represents a single dataset distribution. Therefore, additional experiments and performance evaluations on other datasets are necessary.

        > `ETtv`: The evaluation dataset (ATC) naturally exhibits strong daily regularity, but no experiment tests NeMo-map on non-periodic or transient patterns, making the claimed generality questionable.
    - Addressed by: Extended the evaluation to the ETH/UCY datasets, which contain non-periodic motion patterns in outdoor environments. Included both quantitative and qualitative results in Sec. 4.2 – 4.3. The results demonstrates that NeMo-map can also operate effectively without daily periodic structure. In ETH/UCY, our method achieves lowest NLL across scenes, compare with all baselines. All improvements are statistically significant ($p<0.001$, paired t-tests).

---

> ### Author Response · Authors · 2025-11-28
>
> 2. **Practical downstream evaluation** (Reviewers `9dui`, `xjtj`, `ETtv`)
>     - Reviewer concerns:
>         > `9dui`: Since one of the motivations is to support robot navigation and trajectory prediction, could the authors show how NeMo-map impacts these tasks compared to prior MoD-based representations? It remains unclear whether the proposed NeMo-map actually improves downstream performance in motion planning, collision avoidance, or human trajectory forecasting compared with prior MoD-based methods. Including an evaluation in a planning or prediction pipeline—e.g., comparing robot navigation success rates or forecast accuracy using NeMo vs. CLiFF—would provide stronger evidence of practical benefit.
>
>         > `xjtj`: Given this description, it would be valuable to verify whether, in the field of human motion prediction or human trajectory prediction, the proposed approach indeed provides better priors that contribute to the performance improvements observed in “With MoD guidance” in Figure 1. Such experiments would substantially reinforce the credibility of the claim that MoD guidance effectively aids motion planning and human motion prediction.
>
>         > `ETtv`: The paper reports no task-based or trajectory-level validation (e.g., motion prediction or navigation performance), making it difficult to assess whether the smoother velocity fields actually improve practical planning or prediction outcomes.
>     - Addressed by: Added downstream evaluations on the long-term human motion prediction task, where our approach consistently outperforms MoD-based predictors and state-of-the-art transformer- and diffusion-based baselines. The improvements arise from NeMo-map's ability to capture smooth and precise local flow patterns, resulting in more accurate human motion predictions. This new experiment has been added in Sec. 4.4. Results across multiple prediction horizons (10s–60s) and qualitative prediction examples are included in Fig. 6 of the revised paper.
>
> 3. **Improved qualitative comparison** (Reviewers `9dui`, `xjtj`)
>     - Reviewer concerns:
>         > `9dui`: Could the authors provide qualitative visual comparisons between NeMo-map and the baselines (e.g., CLiFF-map, Online CLiFF-map, STeF-map)? The qualitative results (Fig. 5) present visualizations only for the proposed NeMo-map, without showing side-by-side comparisons with baseline methods such as CLiFF-map, STeF-map, or Online CLiFF-map. This omission makes it difficult for readers to visually assess the claimed advantages in smoothness, continuity, or multimodality. Including qualitative comparisons—e.g., overlaying the predicted flow fields of different models in the same regions—would provide a much clearer and more compelling demonstration of NeMo-map’s improvements.
>
>         > `xjtj`: The authors conducted experiments on the ATC dataset and included the analysis in Section 4.5 (Qualitative Results). However, the explanation could be more detailed. It would be insightful to elaborate on what specific human lifestyle patterns lead to certain behaviors or activities at particular locations and times. Moreover, it would be helpful to clarify under which scenarios(situations) or environmental conditions the model’s predictions are accurate and where they fail. A deeper discussion on these aspects would strengthen the interpretability and practical relevance of the results.
>     - Addressed by: Added multiple representative human motion scenarios and provided detailed comparisons showing how our method models these motion patterns relative to baseline MoDs (Sec. 4.3). Side-by-side visual comparisons between NeMo-map and all baseline methods on both the ATC and ETH/UCY datasets are provided. These comparisons show that NeMo-map:
>          - maintains continuous and coherent flow fields in crossing traffic structures, models multimodal motion patterns consistently (Fig. 3)
>          - adapts to temporal variations in pedestrian motion patterns throughout the day (Fig. 4)
>          - captures emerging directional trends and captures diverse motion patterns in multiple environments (Fig. 5)
>
> 4. **Additional ablations on hyperparameters** (Reviewer `ETtv`)
>     - Reviewer concerns:
>         - `ETtv`: Moreover, the ablation study (Sec. 4.6) isolates temporal encodings but omits critical architecture factors such as mixture component count (J) or grid resolution (Gs), leaving the method’s scalability characteristics underexplored.
>     - Addressed by: Added ablations on SWGMM component number and feature grid resolution (Sec. 4.5, App. E).

---

### Meta-Review · Area_Chair_2rgp · 2026-01-07

**Summary:**

The present paper proposes a new representation for Maps of Dynamics (MoD) for motion planning in mobile robotics. The key idea of MoDs is to encode statistical motion patterns in a map that can then be used for subsequent downstream tasks, such as trajectory prediction or motion planning. The new representation proposed by the authors is continuous and based on implicit neural representations (similar to NeRFs). Their map representation takes coordinates as input and generates the parameters of a Semi-Wrapped (or partially wrapped) Gaussian Mixture Model as output. The authors claim that, compared to existing baselines, their representation achieves better accuracy of motion representation and smoother velocity distributions in sparse regions while still being computationally efficient.

**Reviewer Concerns:**

The reviewers raised several concerns with this work. First, one reviewer (9dui) questioned its conceptual novelty and felt that the work proposes a small improvement over CLiFF-map and a "mere" application of the NeRF idea to maps. Several reviewers criticized the conceptual limitations in assuming temporal periodicity (g9DM, ETtv) and the narrow experimental evaluation focusing on only one dataset, making the generalization capabilities of the model questionable (xjtj). Another concern was the lack of evaluation on downstream tasks (9dui) and the limited interpretability of the representation (xjtj).

**Reviewer Scores:**

The authors addressed the points on evaluation and generalization by providing additional experiments on the ETH and UCY datasets. They also addressed the lack of downstream evaluation by evaluating their idea on the long-term human motion prediction task.

It is laudable that the authors ran additional experiments. Whether these experiments would have satisfied the reviewers is debatable. On the one hand, the authors show usefulness in "less periodic" datasets such as UCY and ETH. On the other hand, these datasets seem to be largely outdated in the motion prediction context. And although they were suggested by one of the reviewers, it is not clear whether all of the reviewers who were critical of the experiments would have been satisfied with this. The authors deserve the benefit of the doubt, but should consider using more recent benchmarks.

Moreover, for the evaluation on the downstream task, the work has chosen a long-term human motion prediction task. Here, it is also unclear to what extent this is sufficient. If the work is positioned as a prediction work, there would have been other baselines to compare against (e.g., Deep Context Maps for a conceptually similar work or some of the WOD Interaction Prediction Challenge contenders for SOTA performance approaches). Again, the authors are given the benefit of the doubt, but should provide some clarification. Finally, the points raised about the conceptual novelty and interpretability were largely not addressed and should be discussed.

---

### Decision · Program_Chairs · 2026-01-26

Accept (Poster)